# Dietary-Lifestyle Patterns Associated with Bone Turnover Markers, and Bone Mineral Density in Adult Male Distance Amateur Runners—A Cross-Sectional Study

**DOI:** 10.3390/nu14102048

**Published:** 2022-05-13

**Authors:** Aleksandra Bykowska-Derda, Magdalena Zielińska-Dawidziak, Magdalena Czlapka-Matyasik

**Affiliations:** 1Department of Human Nutrition and Dietetics, Poznan University of Life Sciences, Wojska Polskiego 31, 60-624 Poznan, Poland; aleksandra.derda@up.poznan.pl; 2Department of Food Biochemistry and Analysis, Poznan University of Life Sciences, Wojska Polskiego 28, 60-637 Poznan, Poland; magdalena.zielinska-dawidziak@up.poznan.pl

**Keywords:** food frequency intake, PINP, cTX, bone loss, fractures

## Abstract

Background: Excessive mileage can be detrimental to bone mineral density among long-distance runners. The negative effects of mileage could be alleviated by appropriate nutrition. The purpose of this study was to analyse the dietary-lifestyle patterns in relation to bone mineral density and bone turnover markers among amateur marathoners. Methods: A total of 53 amateur male distance runners were divided into two clusters by k-means cluster analysis. Bone mineral density was measured by dual X-ray absorptiometry (DXA). Blood was drawn to analyse bone resorption marker C-terminal telopeptide (cTX) and bone formation marker amino-terminal propeptide of type I collagen (PINP). Food frequency intake and lifestyle information were measured by multicomponent questionnaire KomPAN^®^. Yearly average mileage per month was taken from each participant. Results: There were two distinguished clusters: Less-healthy-more-active-low-Z-score (LessHA) (*n* = 33) and More-healthy-less-active-high-Z-score (MoreHLA) (*n* = 20). LessHA had a lower frequency intake of pro-healthy food groups, a lower number of meals during a typical day, and a higher mileage training than the group of athletes who followed the MoreHLA. Athletes following the LessHA pattern also had a lower Z-score in the lumbar spine and femoral bone and a lower PINP. Conclusion: The current study suggests that pro-healthy dietary patterns and lower mileage may favour higher bone mineral density in male amateur marathoners.

## 1. Introduction

Running is one of the most popular and practised sports around the world. The number of participants in running, jogging, and trail running in the U.S. between 2006 and 2017 was almost 60 million [1]. Similarly, the popularity of amateur running in Poland increased threefold between 2018 and 2020 [2]. It is observed that the growing participation in marathon running by amateur, middle-aged cases with a belief that more intense exercise will provide incremental health benefits has become widespread. Running has become a way of spending time actively and caring for one’s physical condition and health.

Despite the health benefits associated with practising amateur jogging, it should be emphasised that practising it without adjusting training loads, supplementation, and diet may be associated with health risks. Amateur marathon runners bear a certain degree of health risks such as sports injuries or nutritional deficiencies. Additionally, experts have warned non-athletic amateurs about the “exercise paradox” and the probable deleterious effects of prolonged high-intensity exercise on the musculoskeletal system [3]. Epidemiological studies suggest a “reverse J shaped” relationship between running intensity and mortality [4]. The greatest benefits of the reduction of cardiovascular mortality and all-cause mortality are achieved at a lower intensity of running, while the benefits tend to be blunted at a higher intensity of running [5,6].

Bearing in mind this data, specialists should have a balanced discussion with amateur runners training for marathons about the risks and benefits of high intensity exercise and should evaluate them to rule out the health risk.

Low bone mineral density increases the risk of osteoporosis and stress fractures [7]. Running, being a resistance exercise, has a positive effect on bone mineral density (BMD) [8,9,10,11]. However, endurance running seems to be less desirable for bone health than sprint disciplines. Elite distance runners tend to have a bone mineral density similar to the non-exercising population and a significantly lower bone mineral density than sprinters [8,9,10]. Excessive mileage with high intensity and inadequate nutrition may be additionally detrimental to BMD, which has been shown on the lumbar and femoral neck of the population of female runners with female athlete triad [12,13].

An effect of energy deficiency similar to female athlete triad can be seen in male athletes, which has been described as Relative Energy Deficiency in Sports Syndrome (RED-S) [14]. The incidence of RED-S is difficult to assess and requires further studies [13]. There is some evidence that shows the BMD of male amateur runners [15]; however, no studies have compared the BMD of amateur runners with their mileage and nutrition behaviours. There are more studies needed concerning male athlete bone health, which is overlooked in sports medicine. The study of young male endurance runners compared to other athletes has shown that lower expectant weight, low BMI, consuming less than one serving of calcium-rich foods, and a history of stress fractures can influence spine BMD among young male runners [16]. Athletes in this experiment were at the age when the skeletal system is not fully matured.

Endurance running is a growing discipline among amateurs who often race multiple marathons during the year and are at an older age in comparison to professional runners who usually race marathons twice a year. Amateur marathon runners, except for the strenuous training regime, also work full time very often, which may leave less time for rest and meal planning. Furthermore, running a marathon is often one of the main goals of a person who wishes to start a healthy lifestyle, often with the addition of a restrictive diet for weight loss purposes. Some studies show that amateur runners have healthier eating habits than non-exercising individuals [17], which in some cases may be the basis of restricted energy intake [12,13,14]. This, in turn, has been associated with low BMD among professional athletes.

To date, there have been no studies that evaluated the factors that influence the bone density of amateur runners, and there are no specific guidelines on nutritional recommendations to avoid the incidence of stress fractures and low bone mineral density.

The purpose of this study is to analyse the impact of diet quality and mileage on bone turnover markers and bone mineral density among amateur marathon runners.

## 2. Materials and Methods

### 2.1. Participant’s Characteristics and Study Design

Healthy male amateur runners from Poland, Wielkopolska region (*n* = 53) were recruited into the study between October and March 2017–2018 from local social media groups and marathon races (Figure 1). During the one laboratory visit, the participants were measured for body composition and bone mineral density, and blood was drawn. They filled out a questionnaire concerning nutritional habits and food frequency intake. According to the inclusion criteria, all of the participants did not take any medication or high-dose vitamin or mineral supplements, trained long distance running for a minimum of 3 years, had to run at least one marathon, and worked full time. The study was accepted by the local IRB board (Poznan University of Medical Sciences, obtained on 7 September 2017, Resolution No. 895/17).

### 2.2. Dietary-Lifestyle Patterns

Dietary-lifestyle patterns (DLPs) were derived using cluster analysis. The input variables were three dietary and four bone mineral density and turnover markers (in g/cm^2^ and ng/mL) and one activity marker (in km/month) component of DLPs. All input variables were standardised to achieve a mean equal to 0 and a standard deviation equal to 1. To identify the optimal number of clusters, the analysis was performed several times. The K-means clustering algorithm was used, and participants were grouped according to Euclidean distances. Finally, two clusters representing two dietary-lifestyle patterns were selected (Figure 2), the first characterised by less-healthy-more-active-lower-Z-score (LessHA) and the second characterised by more-healthy-less-active-higher-Z-score (MoreHLA) dietary-lifestyle patterns.

### 2.3. Nutritional Habits and Food Frequency Intake

To assess the nutritional habits, the validated Dietary Habits and Nutrition Beliefs Questionnaire KomPAN^®^ (The Committee of Human Nutrition, Polish Academy of Science) was used [18]. A total of 24 groups of products were included in the food frequency part of the questionnaire. To estimate the intake, the frequency of intake was calculated into numeric values per day according to previous studies. The numeric values were further analysed as two diet quality scores: the pro-healthy diet index (pHDI-10) and the non-healthy diet index (nHDI-14) previously published and validated [19,20,21,22].

### 2.4. Bone Mineral Density Measurements

All study participants had a dual-energy X-ray absorptiometry (DXA) scan at the Department of Human Nutrition and Dietetics by the same technician on a GE Lunar Prodigy machine (General Electrics Healthcare Medical Systems, Europe, Belgium).

The DXA measured bone mineral content (BMC, g), bone mineral density (BMD, g/cm^2^), and body composition (fat mass, g; fat mass, %; lean mass, g; lean mass, %).

The BMD Z-scores were calculated using the DXA software (Enco) to compare BMD to the average for a healthy person of the same age and sex. The standards of the International Society for Clinical Densitometry (ISCD) recommend using Z-scores to assess BMD in paediatric participants, premenopausal women, and men under 50 years of age [23] (ISCD 2013a, b). The trabecular bone score (TBS) was performed by the same trained researcher using the TBS iNsight software (Medimaps, research version 3.0, Pessac, France). The calculation was performed in the lumbar spine region [24]. The quality controls were performed every day for the DXA equipment to verify system stability.

### 2.5. Bone Turnover Markers and Vitamin D

Blood was taken to assess the serum bone turnover markers C-terminal telopeptide (cTX) and amino-terminal propeptide of type I collagen (PINP), as well as vitamin D status (25-OHD_3_) and blood morphology. The collection took place in the morning in the fasted state, and the participants were advised to not exercise vigorously before the test. All of the material was handled by a certified diagnostic laboratory Synevo sp. z.o.o. Poland. Bone turnover markers were assessed by using the chemiluminescence method. The two kits were used according to the manufacturers’ instructions: IDS-iSYS Intact PINP and IDS-iSYS CTX-I (CrossLaps^®^) (Immunodiagnostic Systems Limited, Boldon United Kingdom). 25-OHD_3_ was measured by the commercially available ELISA kit (BIOHIT OYJ, Helsinki, Finland).

### 2.6. Statistical Analysis

Descriptive statistics (mean, standard deviation median confidence intervals and percentages) were used to characterise the sample. To determine the distribution, the Shapiro–Wilk normality test was used. For the categorised data, the chi^2^ test was used to describe the differences between the categories. To indicate the group with the highest risk of low bone mineral density, k-means cluster analysis was performed. All of the results were standardised before the k-cluster analysis. Statistical analysis, including the k-means cluster analysis *t*-test and Mann–Whitney, was performed in Statistica 13.1 (Stat Soft) software. Bone mineral density Z-scores and T-scores were calculated by the DXA Lunar Prodigy software (GE Healthcare^©^) for each participant.

## 3. Results

### 3.1. Dietary Lifestyle-Patterns

The k-mean cluster analysis concluded with two clusters: Less-healthy-more-active-low-Z-score (LessHA, *n* = 33) and More-healthy-less-active-high-Z-score (MoreHLA, *n* = 20) (Figure 2). The analysis of variance of the k-means clustering is presented in Table 1. The LessHA had a lower frequency intake of pro-healthy food groups, a lower number of meals during a typical day, and a higher mileage training than the group of athletes who followed the MoreHLA cluster. Athletes following the LessHA pattern had a lower lumbar spine Z-score, femoral bone Z-score, and PINP.

### 3.2. Participants’ Characteristics

The participants were amateur marathon runners who ran an average 227 ± 92 km a month, and 75% of them had been practising long-distance running for more than 5 years (Table 2). A total of 70% of the group had higher education, and all of them self-reported their financial status as being average or above average. There were no significant differences between the two groups in terms of BMI, body fat measurements, education level, and financial status. However, there were differences in terms of years of running history, mileage, and the age of the participants between the two clusters.

### 3.3. Frequency of Food Intake

Except for the overall lower pro-healthy diet index (p-HDI) among LessHA, this group had a statistically lower frequency intake of butter and fresh cheese products but a higher intake of vegetables than the MoreHLA group (Table 3).

### 3.4. Bone Turnover Markers

Besides the low lumbar and femoral Z-scores and PINP which were used to distinguish clusters, the LessHA group had lower bone mineral density in the femoral neck, ward’s triangle, and trochanter area of the femoral bone as well as TBS score (Table 4). There were no significant differences between the two groups in terms of total BMD, cTX, and vitamin 25OH-D3.

## 4. Discussion

This study reached its aim by distinguishing the group of amateur marathon runners at a higher risk for low bone mineral density and potential future bone fracture. The low bone mineral density was related to the dietary behaviours and training volume, which confirms the results of the previous studies conducted on professional marathoners [9,13]. There are multiple pieces of evidence that energy intake, the appropriate calcium to phosphorous ratio, family history, high BMI, and weight-bearing exercise influence the bone mineral density in the general population [23,25,26,27,28]. The distinguished clusters differed in bone mineral density at the femoral and lumbar spine sites, PINP bone turnover markers, as well as the intake of pro-healthy foods and average monthly mileage. There was no difference, however, between the two clusters in terms of the frequency intake of non-healthy products, total bone mineral density, cTX bone turnover markers, BMI, or body fat percentage.

### 4.1. Bone Mineral Density

The cluster of runners with lower bone mineral density was characterised by a low total hip Z-score and total lumbar spine Z-score. Lower bone mineral density is related to future fracture risks among adult populations and trained individuals [29]. The TBS has also been lower in this group, and previous studies suggest that this parameter of bone microarchitecture is related to fractures among athletes and that runners seem to have lower TBS values than non-runners [30].

### 4.2. Bone Turnover Markers

The PINP was also statistically lower in the LessHA cluster than in the high BMD cluster, and its levels reflect the number of newly formed collagen molecules [31]. It has been reported that lower levels of PINP are associated with hyperinsulinemia and hyperglycemia in healthy individuals [32]. However, keeping in mind that athletes with statistically lower PINP have higher mileage, the low PINP is related to bone turnover in this case.

The cTX was not statistically different between the two clusters. This bone turnover marker is related to bone resorption, and some studies suggest that it is one of the least influenced by intensive physical exercise bone turnover markers [3,33]. On the contrary, there are studies that show that aerobic exercise bouts increase cTX in male triathletes [34].

### 4.3. Nutrition and Bone Health

The analysis of a marathon runner’s lifestyle factors influencing the BMD and TBS will enable them to take appropriate preventative action.

The runners from the LessHA cluster consumed less frequent meals during a typical day than the runners from the MoreHLA cluster. The high frequency of meals has been associated with a lower body fat and a high fat-free mass among the adult population [35]. Additionally, a high frequency of meals among athletes could provide optimal energy intake [36].

The pro-healthy dietary index (pHDI-10), which was higher among men in the MoreHLA group, includes groups such as milk, fermented drinks, and cottage cheese, among others. High pHDI-10 could contain a substantial number of dairy products. There was a significant difference in the intake of cottage cheese, which is not only a source of calcium but also protein. The LessHA group also has a lower intake of butter. Low protein and fat intake could be one of the causes of the previously mentioned Relative Energy Deficiency in Sports Syndrome (RED-S) [13]. Our results support other studies showing that appropriate nutrition may effectively support bone health not only in female but also in male athletes [29,37,38].

Also surprising is the fact that the runners from the LessHA cluster had a higher frequency intake of vegetables. A high intake of vegetables could indicate that runners could also restrict their energy intake [39]. This is also supported by the fact that there was no difference in terms of intake of unhealthy products between the analysed clusters.

### 4.4. Training Volume and Bone Health

The LessHA cluster is also characterised by high mileage and running history. This supports the fact that intensive long-distance running compromises the bone mineral density among both professional athletes and amateurs [30]. In turn, 25-OHD_3_ was not associated with LessHA, which supports previous studies on bone health in professional athletes [29].

There is a significant age difference among the analysed clusters, which could be a weakness of the study. It is already known that BMD decreases with age. However, in our study, we used a Z score instead, which is adjusted for age [23]. The higher age in the LessHA cluster could be due to the longer history of running in this group, which indicates even more that the problem with low BMD among long-distance runners may start in late adulthood. Additionally, the small group of recruited runners allowed us to carefully measure bone health in multiple ways (Z-score of the lumbar and femoral area, bone turnover markers, vitamin D, TBS) as well as dietary and training behaviours.

## 5. Conclusions

This study addresses for the first time the bone health concern of amateur distance runners and compares it with dietary lifestyle patterns. Our results indicate that a high training load and low frequency intake of healthy foods could be a problem for amateurs who also have different everyday activities such as work. This study supports the experts’ advice about the “exercise paradox,” by which prolonged and intensive physical activity could have a detrimental influence on bone health in amateur marathoners. Additionally, it points out that lifestyle and nutrition are also important factors in male athletes’ bone health, and this matter should be further studied.

## Figures and Tables

**Figure 1 nutrients-14-02048-f001:**
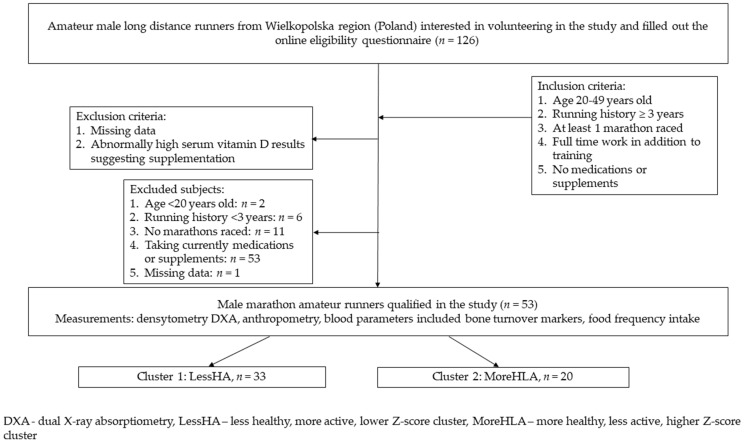
Study recruitment flowchart.

**Figure 2 nutrients-14-02048-f002:**
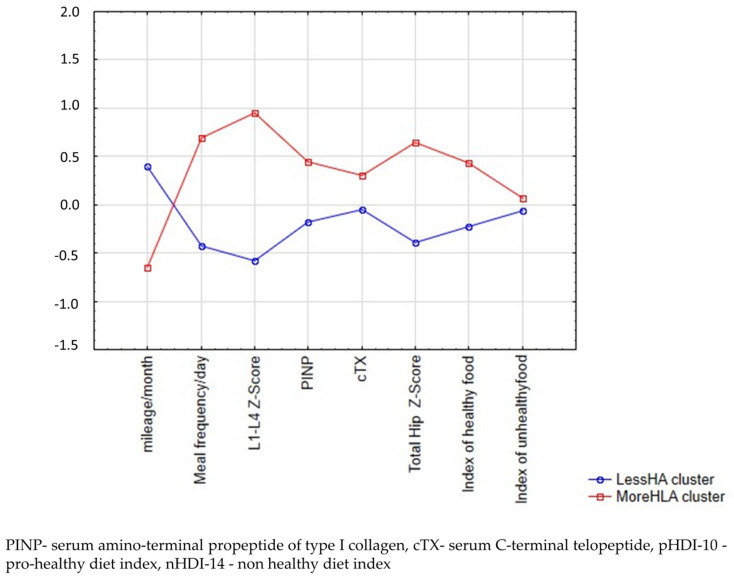
Mean plot of two clusters of male amateur marathoners.

**Table 1 nutrients-14-02048-t001:** The analysis of variance statistics of the K-means clustering of the study group of marathoners.

	SS-between	df	SS-within	df	F	*p*
Mileage/month	13.70	1	41.68	51	16.77	0.00
Meal frequency/day	15.70	1	40.79	51	19.63	0.00
Lumbar L1-L4 Z-Score	29.13	1	23.85	51	62.29	0.00
PINP ^1^	4.89	1	51.64	51	4.83	0.03
cTX ^2^	1.48	1	44.17	51	1.70	0.20
Femoral Z-Score	13.44	1	38.56	51	17.78	0.00
pHDI-10 ^3^	5.34	1	46.19	51	5.90	0.02
nHDI-14 ^4^	0.19	1	52.58	51	0.19	0.67

^1^ PINP—serum amino-terminal propeptide of type I collagen, ^2^ cTX—serum C-terminal telopeptide, ^3^ pHDI-10—pro-healthy diet index, ^4^ nHDI-14—non healthy diet index.

**Table 2 nutrients-14-02048-t002:** Characteristics of the total study sample and comparisons between clusters.

Characteristics	Less-Healthy-More Active-Low-Z-Score (*n* = 33)	More-Healthy-Less-Active-High-Z-Score (*n* = 20)	Total Sample (*n* = 53)	*p*
Age (years):	40.82 ± 5.80	36.65 ± 7.21	39.07 ± 6.32	
Mean ± SD	42.00 (38.64; 42.57)	37.00 (33.22; 39.88)	41.00 (37.30; 40.84)	
Median (CI 95%) <30 *n* (%)	2 (6)	4 (20)	6 (11)	0.04 *
30–34	2 (6)	4 (20)	6 (11)	
35–39	7 (21)	4 (20)	11 (21)	0.14
40–44	13 (39)	3 (15)	16 (30)	
≥45	9 (27)	5 (25)	14 (26)	
BMI (kg/m^2^)	23.98 ± 2.27	24.58 ± 2.29	24.21 ± 2.27	0.41
Mean ± SD	23.60 (23.17; 24.78)	24.30 (23.51; 25.66)	23.80 (23.58; 24.80)	
Median (CI 95%) <25	21 (64)	14 (70)	35 (66)	0.64
≥25	12 (36)	6 (30)	18 (34)	
Fat percentage ^1^	21.32 ± 4.04	20.15 ± 4.06	21.00 ± 4.07	0.39
Mean ± SD	21.38 (19.60; 23.04)	19.97 (17.98; 22.33)	20.87 (19.57; 22.20)	
Median (CI 95%) Athletes (6–13%)	2 (6)	2 (10)	4 (7)	
Fitness (14–17%)	7 (21)	2 (10)	9 (17)	0.73
Average (18–24%)	17 (51)	11 (55)	28 (53)	
Obese (≥25%)	7 (21)	5 (25)	12 (23)	
Education level				
Lower secondary	2 (6)	1 (5)	3 (6)	
Upper secondary	10 (30)	3 (15)	13 (24)	0.34
Higher (e.g., B.S., M.S.)	21 (64)	16 (80)	37 (70)	
Physical activity at work/school (self-reported)				
Low	15 (45)	10 (50)	25 (47)	
Medium	7 (21)	5 (25)	12 (23)	0.81
High	11 (33)	5 (25)	26 (49)	
Financial situation (self-reported)				
Below average	0	0	0	
Average	21 (64)	7 (35)	28 (53)	0.06
Above average	12 (36)	13 (65)	25 (47)	
Average yearly mileage/month				
Mean ± SD	263 ± 89	169 ± 66	227 ± 92	
Median (CI 95%)	250 (231.43; 294.64)	165 (137.89; 199.60)	200 (201.88; 253.02)	
<100 km	0	3 (15)	3 (6)	
100–150 km	4 (12)	5 (25)	9 (17)	0.00 *
151–200 km	8 (24)	7 (35)	15 (28)	
201–250 km	6 (18)	4 (20)	10 (19)	0.00 *
251–300 km	5 (15)	1 (5)	6 (11)	
301–350 km	3 (9)	0	3 (6)	
>350 km	7 (21)	0	7 (13)	
Running history				
Less than 5 years	5 (15)	8 (40)	13 (24)	0.04 *
More than 5 years	28 (84)	12 (60)	40 (75)	

^1^ According to American Council on Exercise; The *p* values below the threshold of statistical significance are marked with the * *p* < 0.05 analysed by the Mann–Whitney test, (continuous variables) or chi^2^ test (categorical variables).

**Table 3 nutrients-14-02048-t003:** The differences in food frequency intake per day between the Less-healthy-more-active-low-Z-score and More-healthy-less-active-high-Z-score clusters of runners.

	Less-Healthy-More Active-Low-Z-Score					More-Healthy-Less-Active-High-Z-Score					*p*
	Mean	±SD	Median	CI (95%)		Mean	±SD	Median	CI (95%)		
White bread	0.83	0.73	0.50	0.57	1.08	0.92	0.69	0.50	0.60	1.24	0.64
Wholemeal bread	0.66	0.65	0.50	0.43	0.88	0.76	0.71	0.50	0.43	1.10	0.57
White rice, pasta, fine-ground groats	0.38	0.24	0.50	0.30	0.47	0.50	0.41	0.50	0.31	0.70	0.19
Buckwheat, oats, wholegrain pasta, or other coarse-ground groats	0.36	0.33	0.14	0.24	0.48	0.55	0.38	0.50	0.37	0.73	0.06
Fast food	0.07	0.08	0.06	0.04	0.10	0.07	0.04	0.06	0.06	0.09	0.96
Fried dishes	0.38	0.25	0.50	0.29	0.46	0.37	0.25	0.50	0.25	0.48	0.90
Butter	0.58	0.70	0.50	0.33	0.83	1.03	0.87	1.00	0.63	1.44	0.04 *
Lard	0.02	0.03	0.00	0.00	0.03	0.01	0.02	0.00	0.00	0.02	0.71
Vegetable oils	0.39	0.38	0.50	0.25	0.52	0.25	0.46	0.06	0.03	0.47	0.25
Milk	0.77	0.48	1.00	0.60	0.94	0.80	0.65	1.00	0.50	1.10	0.86
Fermented milk products	0.42	0.36	0.50	0.29	0.55	0.53	0.33	0.50	0.37	0.68	0.28
Fresh cheese curd products	0.30	0.27	0.14	0.21	0.40	0.51	0.34	0.50	0.35	0.67	0.02 *
Hard cheese	0.51	0.44	0.50	0.36	0.67	0.56	0.34	0.50	0.40	0.72	0.67
Cured meat, smoked sausages	0.68	0.68	0.50	0.44	0.92	0.51	0.50	0.50	0.28	0.74	0.34
Red meat	0.31	0.36	0.14	0.18	0.44	0.32	0.26	0.32	0.20	0.44	0.92
White meat	0.38	0.18	0.50	0.32	0.45	0.43	0.26	0.50	0.31	0.56	0.42
Fish	0.14	0.12	0.14	0.09	0.18	0.19	0.16	0.14	0.12	0.27	0.16
Eggs	0.43	0.39	0.50	0.29	0.57	0.50	0.26	0.50	0.38	0.62	0.51
Pulses-based foods	0.14	0.16	0.06	0.08	0.19	0.17	0.24	0.10	0.06	0.28	0.49
Potatoes	0.28	0.19	0.14	0.21	0.35	0.27	0.20	0.14	0.17	0.36	0.77
Fruit	0.98	0.64	1.00	0.75	1.20	1.13	0.56	1.00	0.86	1.39	0.40
Vegetables	0.85	0.56	0.50	0.65	1.05	1.23	0.68	1.00	0.91	1.55	0.03 *
Sweets	0.57	0.48	0.50	0.40	0.74	0.60	0.41	0.50	0.41	0.80	0.80
Instant soups or ready-made soups	0.06	0.15	0.00	0.01	0.11	0.02	0.04	0.00	−0.00	0.04	0.19
Tinned meats	0.02	0.03	0.00	0.01	0.03	0.01	0.02	0.00	−0.00	0.01	0.04 *
Tinned vegetables	0.18	0.18	0.14	0.11	0.24	0.15	0.19	0.06	0.06	0.23	0.57
Fruit juices	0.32	0.27	0.14	0.22	0.41	0.33	0.30	0.14	0.18	0.47	0.91
Vegetable juices	0.18	0.23	0.06	0.10	0.26	0.31	0.31	0.14	0.17	0.46	0.07
Sweetened hot drinks	0.70	0.85	0.14	0.39	1.00	0.52	0.88	0.00	0.11	0.93	0.47
Sweetened drinks	0.08	0.17	0.06	0.02	0.14	0.07	0.11	0.06	0.02	0.13	0.79
Energy drinks	0.79	0.39	1.00	0.65	0.92	0.91	0.28	1.00	0.78	1.04	0.22
Water	1.34	0.79	2.00	1.06	1.62	1.70	0.55	2.00	1.44	1.96	0.08
Alcoholic beverages	0.22	0.24	0.14	0.14	0.31	0.25	0.27	0.10	0.13	0.38	0.70
pHDI-10 ^1^	24.98	9.51	23.10	21.61	28.35	31.54	9.57	30.80	27.06	36.02	0.02 *
n-HDI-14 ^2^	19.31	6.90	18.74	16.86	21.76	20.11	5.64	20.11	17.47	22.74	0.67

The *p* values below the threshold of statistical significance are marked with the * *p* < 0.05 analysed by Mann–Whitney test or *t*-test. ^1^ pHDI-10—pro-healthy diet index, ^2^ nHDI-14—non healthy diet index.

**Table 4 nutrients-14-02048-t004:** Bone mineral density and bone metabolism markers between the two clusters of runners.

	Less-Healthy-More Active-Low-Z-Score					More-Healthy-Less-Active-High-Z-Score					
	Mean	±SD	Median	CI (95%)		Mean	±SD	Median	CI (95%)		*p*
25OH-D3 ^1^ (ng/mL)	30.58	11.27	29.30	26.58	34.58	27.99	8.12	28.15	24.18	31.79	0.49
cTX ^2^ (ng/mL)	0.28	0.11	0.29	0.24	0.32	0.33	0.19	0.26	0.24	0.43	0.58
PINP ^3^ (ng/mL)	61.60	19.58	58.10	54.65	68.54	75.87	27.60	72.20	62.95	88.78	0.05 *
Total BMD ^4^ (g/m^2^)	1.15	0.13	1.14	1.10	1.19	1.11	0.17	1.00	1.03	1.19	0.35
Total Hip T-Score (-)	−0.17	0.80	−0.10	−0.46	0.11	0.85	0.82	0.90	0.47	1.24	0.00 *
Femoral Neck Z-Score (-)	0.10	0.78	0.17	−0.18	0.38	1.04	0.94	1.01	0.60	1.48	0.00 *
Ward’s Triangle Z-Score (-)	−0.30	0.84	−0.15	−0.60	−0.00	0.93	1.32	0.71	0.31	1.55	0.00 *
Trochanter Z-Score (-)	−0.12	1.06	0.04	−0.50	0.25	0.99	0.85	1.04	0.59	1.39	0.00 *
Total Hip Z-Score (-)	0.07	0.78	0.23	−0.21	0.35	0.99	0.73	1.01	0.64	1.33	0.00 *
L1-L4 BMD ^4^ (g/m^2^)	1.15	0.09	1.17	1.11	1.18	1.38	0.12	1.39	1.33	1.44	0.00 *
L1-L4 T-Score (-)	−0.65	0.74	−0.52	−0.91	−0.39	1.34	1.01	1.35	0.87	1.82	0.00 *
L1-L4 Z-Score (-)	−0.58	0.80	−0.56	−0.86	−0.29	1.34	0.94	1.43	0.90	1.78	0.00 *
TBS (-) ^5^	1.38	0.08	1.36	0.06	0.11	1.45	0.08	1.46	0.06	0.12	0.00 *

The *p* values below the threshold of statistical significance are marked with the * *p* < 0.05 analysed by Mann–Whitney test or *t*-test. ^1^ 25OH-D3—serum vitamin D, ^2^ cTX—serum C-terminal telopeptide, ^3^ PINP—serum amino-terminal propeptide of type I collagen, ^4^ BMD—bone mineral density, ^5^ TBS—trabecular bone score.

## Data Availability

The datasets used and analysed during the current study are available from the corresponding author on reasonable request.

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
