# Peer review of "Dietary-Lifestyle Patterns Associated with Bone Turnover Markers, and Bone Mineral Density in Adult Male Distance Amateur Runners—A Cross-Sectional Study"

_nutrients, 2022, doi:10.3390/nu14102048_

Round 1

Reviewer 1 Report

  1. The Inclusion Criteria and Exclusion Criteria

Exclusion criteria are exclusions for individuals who meet the inclusion criteria, not the same as inclusion criteria.

  1. Statistical Description

The mean ± SD is used for the data conforming to the normal distribution; Median/quartile is used for skew data

  1. Presentation of Statistical Results

Only the P-value is too thin; perhaps other analyses can be added to make the results more intuitive and credible

Author Response

Dear Reviewer,

Thank you for revising our manuscript titled “Dietary-lifestyle patterns associated with bone turnover markers, and bone mineral density in adult male distance amateur runners. A cross-sectional study”. We greatly appreciate the time and efforts to review our manuscript and we agree that the proposed changes will contribute to the improvement of our manuscript. We have addressed all issues indicated in the review, and we believe that the revised version can meet the journal publication requirements.

Please find our responses to your comments below.

Rev 1. Comments

Answer/correction

1

 The Inclusion Criteria and Exclusion Criteria:

Exclusion criteria are exclusions for individuals who meet the inclusion criteria, not the same as inclusion criteria.

Thank you for the correction. We corrected the inclusion and exclusion criteria in the Figure 1

2

Statistical Description

The mean ± SD is used for the data conforming to the normal distribution; Median/quartile is used for skew data

Median has been added to the Table 2. Characteristics of the patients.

3

Presentation of Statistical Results

Only the P-value is too thin; perhaps other analyses can be added to make the results more intuitive and credible

Table 1 with analysis of variance has been added to the manuscript which will make the results more intuitive and credible.

Reviewer 2 Report

Aleksandra Bykowska-Derda et al have performed an interesting study aimed to investigate in male running bone condition. As reported by authors, this aspect is well examined in female athletes but less in male. 

The data obtained show how excessive physical activity not adequately supported by a diet rich in healthy foods can damage health even in amateur sportsmen.
The work is well designed and the discussion is consistent with the data obtained. I just have some suggestions:
*The type of test performed to determine food intake should be indicated in the abstract;
*Subject’s characteristics (table 1):  I would also add the type of work performed by subjects, as it is known particularly strenuous work ( shifting of heavy loads or shift work) can negatively affect bone health;
*Discussion: in addition to "exercise paradox", I would underline how this work shows the need to develop gender medicine , including sport gender medicine and nutrition gender medicine. The relationship between sport and bone health is often studied only in women athletes and is wrong as analyzed by Aleksandra Bykowska-Derda et al.

Author Response

Dear Reviewer,

Thank you for revising our manuscript titled “Dietary-lifestyle patterns associated with bone turnover markers, and bone mineral density in adult male dis-tance amateur runners. A cross-sectional study”.

We greatly appreciate the time and efforts to review our manuscript and we agree that the proposed changes will contribute to the improvement of our manuscript. We have addressed all issues indicated in the review, and we believe that the revised version can meet the journal publication requirements. Please find our responses to your comments below.

Rev 2. Comments

1

The type of test performed to determine food intake should be indicated in the abstract;

We added information about the KomPAN questionnaire in the abstract.

2

Subject’s characteristics (table 1):  I would also add the type of work performed by subjects, as it is known particularly strenuous work ( shifting of heavy loads or shift work) can negatively affect bone health;

Thank you very much for the suggestion. We have this information included in the questionnaire. We added it in the Table 1.

3

Discussion: in addition to "exercise paradox", I would underline how this work shows the need to develop gender medicine , including sport gender medicine and nutrition gender medicine. The relationship between sport and bone health is often studied only in women athletes and is wrong as analyzed by Aleksandra Bykowska-Derda et al

We added this important information in Introduction v70-73, Discussion v.265-267 and in the Conclusions section v. 295-297